# Environmental Noise Impact Assessment for Large-Scale Surface Mining Operations in Serbia

Uros Pantelic * , Petar Lilic, Aleksandar Cvjetic  and Nikola Lilic

Faculty of Mining and Geology, University of Belgrade, Djusina 7, 11000 Belgrade, Serbia
* Correspondence: uros.pantelic@rgf.bg.ac.rs; Tel.: +381-64-6161644

**Abstract:** Noise emissions are a significant environmental impact caused by the mining industry in all technological phases of surface mining, mineral processing, and waste disposal. This paper presents the role of noise impact assessment and control in large-scale surface mining operations. Mine planning develops the model of mining operations, ore excavation, and waste dumping scheduling and processing rates, including spatial distribution of mining activities. Such a level of mine planning requires an environmental impact assessment study. This can be achieved by applying noise impact assessment models. The described approach can be used to verify the effectiveness of the proposed protection measures to reduce or eliminate the identified negative impacts. This paper presents a case study of environmental noise impact assessment and control at the Serbia Zijin Copper DOO Bor mine, encompassing the analysis of the noise protection measures efficiency within the planning of large-scale mining operations at the open-pit mine Veliki Krivelj.

**Keywords:** mine planning; noise impact assessment; mining; noise mapping; environmental impact assessment

## 1. Introduction

Surface mining and mineral processing, with all their characteristics, usually potentially represent a threat to the environment. Therefore, activities such as exploration, planning, excavation, crushing, milling, flotation, and waste disposal appear as potential sources of problems in the field of environmental protection. In modern mining theory and practice, large open-pit mines and the high intensity of mining operations present a great challenge [1–4].

Noise in the mining industry is one of the more prevalent environmental issues. Drilling rigs, loaders, trucks, bulldozers, crushers, mills, screens, and other frequently used equipment in surface mining, as well as blasting, are noisy by nature. Consequently, noise has long been widely recognized as one of the risks to employees in the mining operations environment [5,6]. In circumstances when open-pit mines are in the immediate vicinity of a residential object, as it is in this case, the exposed population could be larger, and the hazard is not always limited to employees with all its consequences.

The most common negative health effects related to prolonged exposition to noise are sleeping disorders with awakenings [7,8], learning impairment [9,10], hypertension, and ischemic heart disease [11–13]. In order to prevent such effects, the European Commission's END (Environmental Noise Directive) was published in 2002, requiring noise maps and action plans every 5 years for the major sources resulted to be the most impactful on human lifestyle: road traffic [14–16], railway traffic [17,18], airports [19,20], and port activities [21,22] are the most diffused ones. Industries should comply with the requirements and mining facility would fall into this category despite the nature of mining environmental noise emissions, for example, time-varying noise levels, dominant low frequency, irregular loud impulsive sounds, etc., which is slightly different from that of traffic (road, railway) and

airports and port activities. In this case study noise assessment will be performed following the national criteria and noise standards [23–28].

Noise in open-pit mines has been recognized as a research subject by many authors [29–35]. Based on a large amount of data from monitoring stations for noise propagation in the open-pit mine Tuncbilek (Turkey) the authors Sensogut and Cinar [30,31] developed an empirical model for calculating the distribution of noise from different sources at open-pit mines. Pathak, Durucan, and Kunimatsu [32] developed a technique for predicting noise levels caused by the operation of a specific group of mining machines. Using this noise assessment, a comprehensive sound field forecast in the vicinity of a surface mine can be made. In order to predict far-field noise levels of the specific set of mining machinery, Nanda, Tripathy, and Patra [35] developed fuzzy inference system-based noise prediction models. The research by Lilic et al. [29,33,34] concerning open-pit mines resulted in the development of a noise mapping model with the aim of defining measures for reducing the negative noise impacts in the immediate vicinity of open-pit mines.

From a long-term perspective, the life cycle of a mine is decided by a number of factors, which is why its development takes place in phases, according to a certain dynamic. Each phase is characterized by the appropriate degree of equipment engagement and the location and number of work sites. The equipment is mostly the same during all phases, but its number and the location of engagement can change significantly depending on the mine development phase, as well as on the work dynamics within the same phase. Such situations hamper effective and efficient decision-making at a given moment. It means that the analysis should include as many potential situations as possible in order to have a timely and adequate response to each of them. The noise mapping process can provide a possible solution in these situations. In addition to its efficiency, noise mapping is also a relatively inexpensive approach to solving environmental noise issues. When the acoustic model is formed, the biggest advantage of this approach is an almost unlimited number of scenarios—potential situations that can become reality at any given time, and for which the best solutions in terms of environmental noise control can be defined in advance, on a daily basis for operational purposes or on strategic bases for mine planning and designing. For this purpose, a noise mapping package [36] was used to simulate possible scenarios of sound-pressure level distributions around the open-pit mine for identified sound sources.

Noise emission presents a significant impact of the mining industry on the quality of the environment in all technological phases of surface mining, mineral processing, and waste disposal. A modern approach to noise emission management at mines includes understanding the types of sources, adhering to the effective modern methods of protection, application of experience, and best noise management practices to bring noise levels down to below the maximum allowed values [29].

Noise emission management in mine planning is a complex procedure due to the large number of factors that affect environmental noise propagation. Modern noise management includes the application of a noise propagation estimation model and an approach based on the global information system (GIS) platform. This method provides access to relevant information needed to take the measures necessary to organize the protection against excessive noise. Generally, the management of environmental quality requires an interdisciplinary approach, that is, it requires a system-oriented approach based on functional abstraction rather than structural decomposition [37].

Noise impact identification, assessment, and control are part of an integral and comprehensive impact assessment and management process, with a large number of interactive and competitive activities. Modern trends in environmental management usually rely on a generic procedure defined by standards within the ISO 14000 group—Environmental Management [28]. Applying a theoretical approach in the process of integrating environmental impact assessment (EIA) and environmental management systems (EMS) is also recommended [38–41].

The aim of this research is to explore and demonstrate the possibility to integrate EIA and EMS in an actual mining project. In line with the integrated systems approach to impact

assessment and environmental management, both tools can provide a focus on significant impacts, identifying them at an early stage, that is, in the project planning phase [38]. This paper presents an approach to environmental noise impact assessment and control in the Serbian mine planning theory and practice. A case study of noise management at the Serbia Zijin Copper DOO Bor mine is presented, including an analysis of the efficiency of planned protection measures from noise within the long-term planning of copper ore mining and processing at the mine.

## 2. Materials and Methods

### 2.1. The Conceptual Framework

The conceptual framework for the integration of noise impact assessment and control in the planning phase of the environmental management system is shown in Figure 1 [29,42,43].

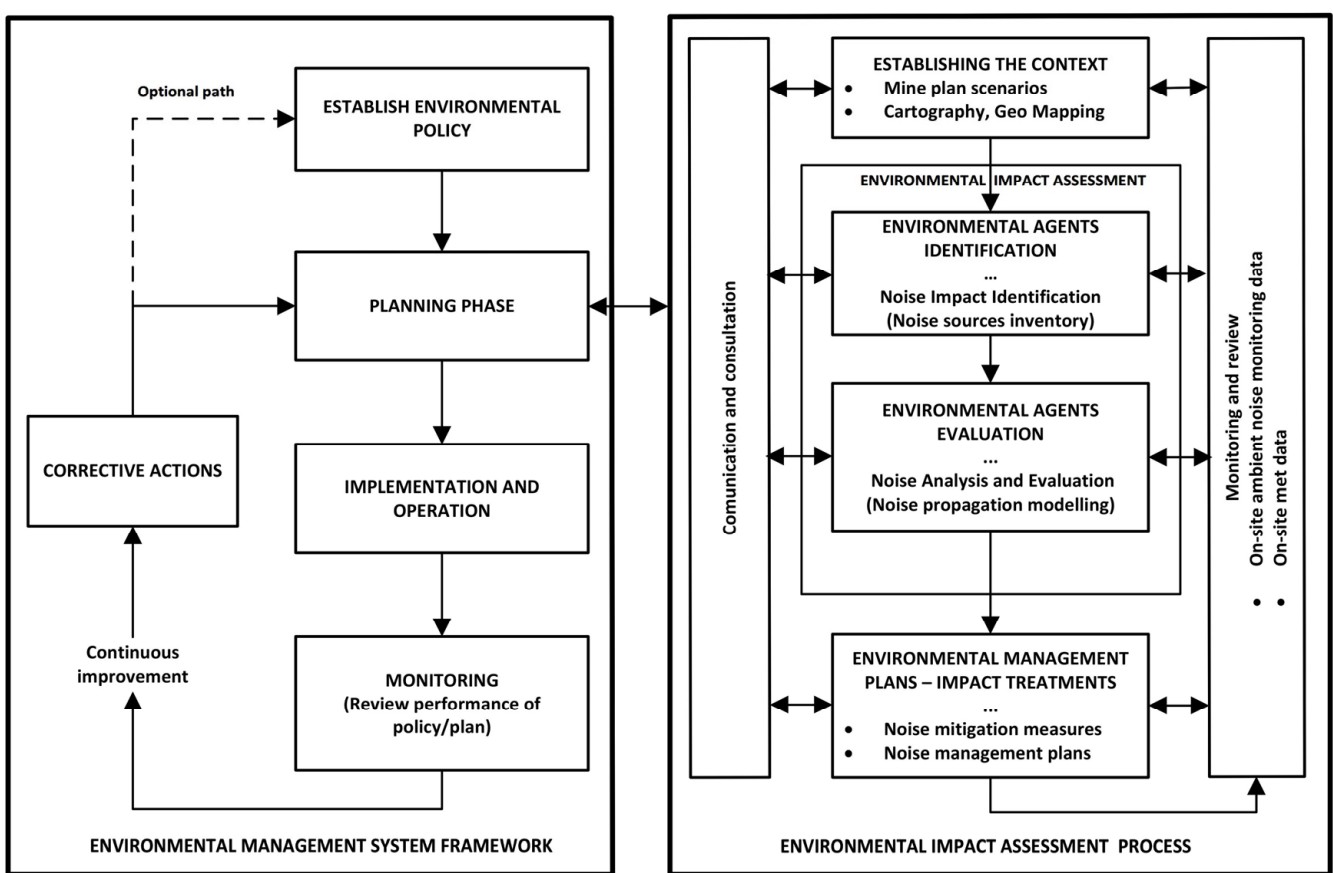

**Figure 1.** The conceptual framework for the identification, assessment, and control of noise as an environmental aspect and its place in the environmental management system [29,42,43].

The aim of the EIA process, as it is shown in Figure 1, is to anticipate and mitigate the impacts of a mining project at an early stage in the project planning phase [38]. The EMS enables managing the environmental impact that occurs on a daily basis during the development and operation of mining activities. In accordance with the system approach to impact identification and assessment, using EIA and EMS could provide focus on significant impacts thus facilitating their identification at an early stage in long and short-term mine planning [38]. The conceptual framework for the integration of noise impact assessment and control in the planning phase of the EMS is shown in Figure 1 [29,42,43].

The mine planning phase, with its objectives and processes, develops the model of mining operations, ore excavation, and waste dumping scheduling and processing rates, including spatial distribution of mining activities. All the objectives and processes of the

mine planning phase must deliver results complying with the organization's environmental policy regarding the organization's EMS. The noise impact assessment and control involved in the EIA process of the planned mining activities should assure their compliance with the organization's environmental policy.

### 2.2. Noise Assessment

Once the baseline information is available, and the likely changes in the environment caused by the project development are recognized through impact prediction, the next stage in the EIA process is impact assessment. The noise assessment generally involves the assessment of the identified noise impacts. This requires interpretation of the importance or significance of the impacts to provide a conclusion, which can ultimately be used by decision makers in determining the fate of the project application.

A noise assessment typically involves the following processes:

1. Identification of the types and number of noise sources.
2. Identification of the representative locations on the mining site. Typically, for an open-pit mine, recognition of the various development phases of interest related to the open-pit size, depth, and source locations.
3. Identification of the seasonal or typical meteorological conditions at day, evening, and night periods.
4. Noise source noise emission measurements and determination of representative sound power levels and frequency spectra for each of the sources, stationary and moving.
5. Number and location of noise-sensitive receptors up to 5 km distant from the mine site. For very quiet background noise levels at night, even greater distances may be required.
6. Selection of a noise modeling software, and specifically how the meteorological variations will be considered and accounted for in the noise emission predictions. Modeling software is used to generate noise contours (noise maps) for the sensitive receiver areas surrounding a mine site and included haul routes. One of the features of noise prediction software is the ranking of noise source contributions at each receiver for each modeled scenario. This is the most important step in the noise assessment and potential mitigation process because it identifies the noise sources which are dominant at each receiver, and which must be controlled if compliance with the noise criteria is to be achieved by a noise management strategy.

Noise mapping in a certain area is a procedure of assessing the exposure to certain noise levels of all sensitive receptors or objects of interest in a certain area, due to the existence of different noise sources in that area or in its environment [33,34,44]. Noise maps are not usually made according to the sound level measurements but are created using calculations and sophisticated computer modeling software. This is especially important in the phase of action planning—response to noise, and especially when the possibility of cost-benefit analysis of options is invaluable in the process of making a final decision in planning the mining activities.

Almost all commercially available noise modeling software provides acknowledged calculation methods according to the relevant standards. The most common noise propagation calculation method, originating from industrial sources, is defined by ISO 9613-1:1993 and ISO 9613-2:1996 [27]. There are several widely accepted noise models that support the recognized and adopted worldwide standards. For this paper's research purposes, the SoundPlan v.8.1 software package was used [36], with the implementation of the ISO 9613 standard [27]. Generally, the standard describes a method for calculating the attenuation of sound during propagation outdoors in order to predict the levels of environmental noise at a distance from a variety of sources. The method predicts the equivalent continuous A-weighted sound-pressure level (as described in ISO 1996 [27]) under meteorological conditions favorable to propagation from sources of known sound emission.

Input data required in the model are

- topographical data;

- ground absorption;
- source sound power levels;
- meteorological conditions.

Topographical data were based on that provided by Serbia Zijin Copper DOO Bor. The contours are at 5 m intervals and cover the project area.

Ground absorption varies from a value of 0 to 1, with 0 being for an acoustically reflective ground (hard ground: paving, ice, concrete, and all other ground surfaces having a low porosity) and 1 for acoustically absorbent ground (porous ground: grass, trees, or other vegetation and all other ground surfaces suitable for the growth of vegetation, such as farming land). In this instance, a value of 1 has been used. If the surface consists of both hard and porous ground, then G takes on values ranging from 0 to 1. For the circumstances presented in the paper, a value of 0.8 has been used.

All mobile and fixed plants are assumed to be operational. For mobile plant noise, data are based on "high idle" or "high load conditions". In order to simulate haul trucks waiting to be loaded, for the haul trucks operating in the pit, half were based on the above conditions and half on "low idle" conditions. Such a situation represents the worst-case scenario as it assumes all plants are operational. In reality, this is rarely the case, particularly at night, which represents the critical assessment period. The number of significant mobile and fixed noise sources assumed in the modeling is provided in Section 3.1.

SoundPlan (with ISO 9613-2:1996 standard implemented) calculates noise levels for certain meteorological conditions (humidity, air pressure, and temperature). The environmental parameters are important to calculate air absorption. The aforementioned meteorological conditions are for downwind propagation or, equivalently, propagation under a well-developed moderate ground-based temperature inversion, which commonly occurs at night. Inversion conditions over water surfaces are not covered. Downwind propagation conditions for the method specified in ISO 9613-2:1996 are

- wind direction within an angle of $\pm 45°$ of the direction connecting the center of the dominant sound source and the center of the specified receiver region, with the wind blowing from the source to the receiver;
- wind speed between approximately 1 m/s and 5 m/s, measured at a height of 3 m to 11 m above the ground.

The equations for calculating the average downwind sound-pressure level $L_{AT}$ (DW) in this part of ISO 9613 are the average for meteorological conditions within these limits. The term average here means the average over a short time interval. These equations also hold, equivalently, for average propagation under a well-developed moderate ground-based temperature inversion, which commonly occurs on clear, calm nights.

The following meteorological conditions were used for the modeling, the results of which are presented in this paper:

- temperature: 11 °C;
- air humidity: 72%;
- atmospheric pressure: 972 mbar.

As for the specific model in the paper, the model itself does not include noise emissions from any source other than the proposed mining operations. Therefore, noise emissions from any other neighboring noise sources, road traffic, and other extraneous sources are excluded from the modeling.

### 2.3. Overview of the Mining Operations and Baseline Conditions

The noise impact assessment and the choice of environmental protection measures during the planning phase of surface mining and mineral processing will be presented in the case study of the Serbia Zijin Copper DOO Bor mine. After the Chinese company Zijin acquired RTB Bor company and established Serbia Zijin Copper DOO Bor, the new development plan of the open-pit mine Veliki Krivelj predicted the increase of the annual ore processing capacity from the existing capacity of 10.6 million TPA (tons per annum) to

23.1 million TPA of dry ore. An expansion of the open-pit mine is necessary to support the planned ramp-up of production rates, as well as a significant increase in waste removal capacity. The planned quantities of waste in the next twenty years will demand additional space to be allocated for the formation of new waste dumps due to insufficient space in the existing locations (old mine Bor, Todorov Potok, and Istočni Planir).

Production at the Veliki Krivelj open-pit mine is based on drilling and blasting technology. The ore haulage to crushing and flotation facilities is based on dump trucks. The waste is transported to external waste dumps and to the old open pit. The flotation waste, tailings, are disposed of at the Veliki Krivelj tailing ponds. The locations of these mining facilities, included in the long-term production scheduling based on the new optimization of the final Veliki Krivelj open-pit mine contour, are shown in Figure 2.

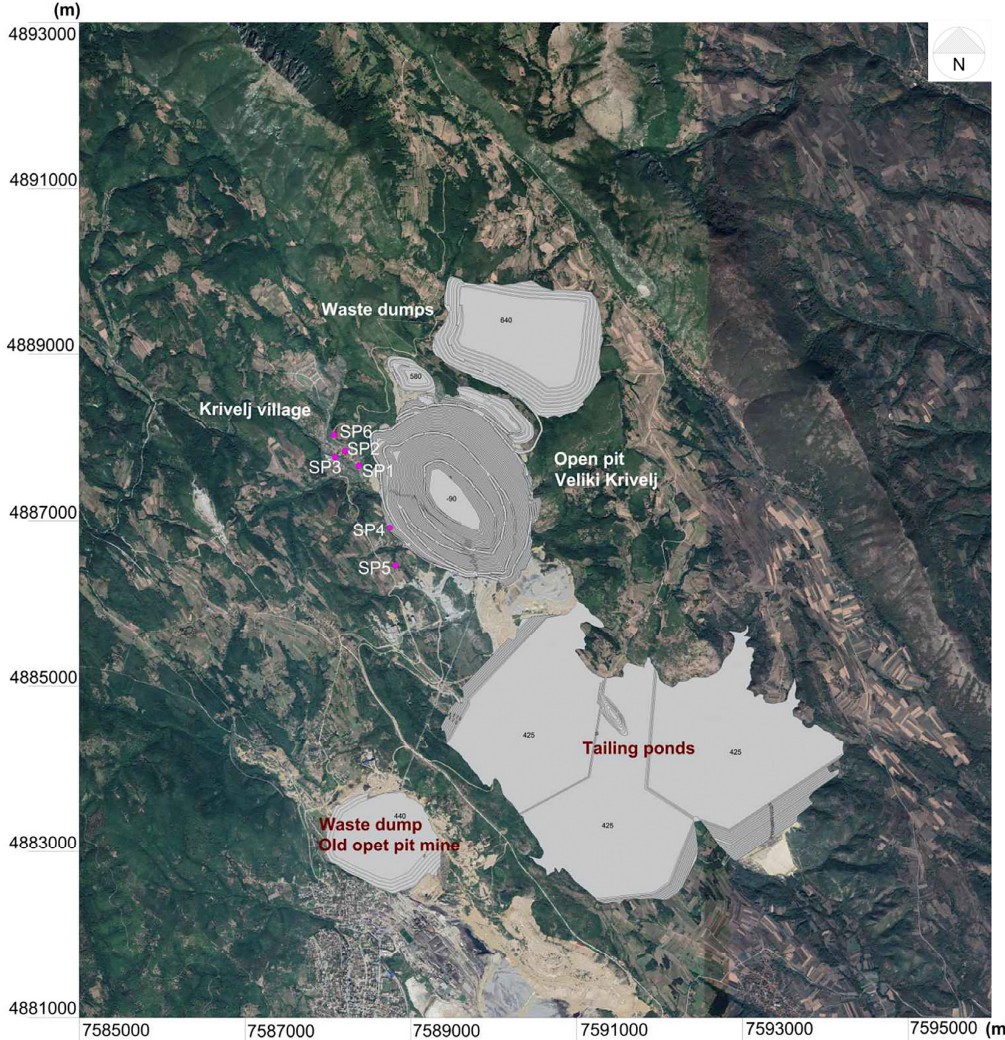

**Figure 2.** Location map of Veliki Krivelj mine facilities (open pit, waste dumps, tailing ponds) with the location of sensitive points SP1–SP6.

From the point of the EIA process, it is essential to collect all relevant information on the current status of the environment as it provides a baseline against which changes due to development can be measured. Environment quality monitoring procedures include the periodical measurements of environmental noise levels in the mine's vicinity, and multiple other parameters. All measurements are performed in accordance with the national legislation that is harmonized with the generally accepted standards ISO 1996-1 and ISO 1996-2 [25,26]. The latest noise levels are shown in Table 1. The measurements were performed at the locations of rural households in Krivelj village, close to the western

boundary of the Veliki Krivelj open pit (sensitive points 4 (SP4), SP5, and SP6: Figure 2 and Table 1).

**Table 1.** The results of noise measurements in the vicinity of the Veliki Krivelj open-pit mine.

| Measurement Location | Rating Level LRAeqT, dB(A) | | | Maximum Permissible Level, dB(A) | |
|---|---|---|---|---|---|
| | Day | Evening | Night | Day and Evening | Night |
| SP4 (House Nikolic) | 39.1 | 38.9 | 37.1 | | |
| SP5 (House Zurkic) | 45.6 | 43.2 | 41.8 | 55 | 45 |
| SP6 (House Karabasevic) | 37.7 | 40.3 | 40.5 | | |

The noise measurement was done by the laboratory accredited by the national accreditation authority ATS (Accreditation Body of Serbia). The noise sampling and analyses on the selected location were performed using the sound analyzer Bruel & Kjaer type 2250. The measurement was in accordance with the standards ISO 1996-1 and ISO 1996-2 [25,26].

According to the national legislation on noise [23,24], the location belongs to acoustic zone 3, which is characterized by purely residential areas and for which the maximum noise levels of 55 dB(A) are prescribed during the day and evening and 45 dB(A) during the night. According to the same legislation, the maximum permissible noise level values relative to the purpose of the premises are 35 dB(A) during the day and evening and 30 dB(A) during the night in living rooms (bedroom and living room) in a residential building with closed windows. As per national legislation, the period of 24 h is divided into three reference time intervals: the day lasts 12 h (from 06:00 to 18:00), the evening last 4 h (from 18:00 to 22:00), and the night lasts 8 h (from 22:00 to 6:00 the next day).

At the given moment, this house was potentially impacted by mining activities at the open-pit mine, which included the operation of the crushing plant: the plant consists of primary and secondary crushers, active 24 h a day with periodic interruptions caused by the dynamics of ore delivery; excavation and loading of ore and waste shovels, which are active 24 h a day, with periodic interruptions caused by the dynamics of ore loading; haulage of ore and waste dump trucks, which move on an unpaved road that connects the open-pit mine with crushers and waste dumps.

At the time of measuring, all the mentioned mechanization and plants were working at full capacity. Based on the measurement of environmental noise levels that originated from mining operations at the open-pit Veliki Krivelj, it can be concluded that at that time the residential building in question was not significantly impacted by the excessive noise levels.

## 3. Results and Discussion

### 3.1. The Assessment and Control of Noise Induced by Common Activities of the Open-Pit Mine

To produce a proper and complete assessment of the potential threat to the nearest residential buildings, the modeling and analysis of the noise propagation result around the open-pit mine Veliki Krivelj has been done for several potential scenarios. This enabled investigation of the planned noise protection measures and their effectiveness in long-term planning. For the research presented in this paper, a typical scenario is selected. The scenario selection criteria were (a) work schedules, i.e., a mine plan/scenario when it is realistic to expect a large number of listed and dominant noise sources in simultaneous work; (b) proximity of mining works to residential buildings in the vicinity of the open-pit mine Veliki Krivelj, mainly in the village of Krivelj, located northwest of the mine boundary (Figure 2).

The following production conditions correspond to the example presented in the paper: activities are in the VI phase of mine development, which corresponds to the 13th year on the production timeline; according to the designed work schedule, the maximum

work capacity is expected this year, which will require the maximum engagement of mining machinery, and as a consequence, the highest level of potentially emitted noise is expected from the open-pit mine Veliki Krivelj; engagement of the following equipment is planned, i.e., noise sources: 4 bulldozers, 3 graders, 7 shovels, 58 dump trucks, 6 drill rigs, primary and secondary ore crusher, waste crusher, belt conveyor for waste, and belt conveyor for ore. What is particularly interesting for this phase of mine development is the spatial relationship between the mine and the residential buildings (the Krivelj village), because production takes place in the immediate vicinity of the northwestern border contour of the mine, which extends in the direction of the Krivelj village, thus making this scenario relevant for the application of noise modeling. Field noise measurements have been carried out extensively to study the noise levels of individual equipment included in the modeling (Table 2).

**Table 2.** Noise levels of mining, auxiliary, and other equipment.

| Noise Source | Source Type | Pcs. | Lw dB(A) | Effective Usage, % of 24 h * | 1/1 Octave Spectrum, dB(A) | | | | | | | |
|---|---|---|---|---|---|---|---|---|---|---|---|---|
| | | | | | 63 Hz | 125 Hz | 250 Hz | 500 Hz | 1 kHz | 2 kHz | 4 kHz | 8 kHz |
| Dump trucks | Line | 58 | 128 | 85 | 119 | 122 | 121 | 121 | 119 | 116 | 109 | 103 |
| Shovels | Point | 7 | 108 | 85 | 100 | 105 | 100 | 95 | 90 | 88 | 84 | 77 |
| Bulldozers | Point, moving | 4 | 125 | 60 | 106 | 110 | 115 | 117 | 121 | 118 | 113 | 109 |
| Drilling rigs | Point | 6 | 113 | 75 | 109 | 108 | 104 | 100 | 105 | 99 | 95 | 86 |
| Graders | Point, moving | 3 | 110 | 60 | 106 | 105 | 101 | 97 | 102 | 96 | 92 | 83 |
| Crushers (ore) | Point | 2 | 130 | 85 | 88 | 99 | 11 | 121 | 125 | 126 | 121 | 119 |
| Crushers (waste) | Point | 1 | 130 | 85 | 88 | 99 | 11 | 121 | 125 | 126 | 121 | 119 |
| Ore conveyor belt | Line | 1 | 122 | 85 | 81 | 91 | 104 | 114 | 117 | 118 | 113 | 111 |
| Waste conveyor belt | Line | 1 | 122 | 85 | 81 | 91 | 104 | 114 | 117 | 118 | 113 | 111 |

* Utilization of mining equipment based on a 24 h period, obtained from the daily production report.

Since the sound power data of the sources were obtained from the sound pressure, measured in the vicinity of the sources, the model was validated. Full model validation of the entire open-pit mine, with all its sources, was performed by comparing the measured and calculated values on several receivers, as shown in Figure 3.

The model validation demonstrates that the noise model is a reasonable representation of the present state of mining activities at the open-pit mine Veliki Krivelj (current ore production rate of 10.6 million TPA). The differences between the measured and calculated values of sound-pressure levels are slightly above the smallest perceptible change of 1–2 dB(A).

According to the previously mentioned environmental noise measurements, the nearest residential building in the vicinity of the open-pit mine Veliki Krivelj is a secluded rural household near the southern entrance to the Krivelj village (sensitive point 4, SP4). At the time this house was located about 550 m from the southwestern rim of the mine and about 750 m from the hauling road for dump trucks. Given that the mine borders will be expanded, Figure 2 shows the selected situation in the mine used for model development as well as the location of individual facilities. Sensitive points SP1, SP2, and SP3 are located near certain facilities in the nearest village (Figure 2, Table 3).

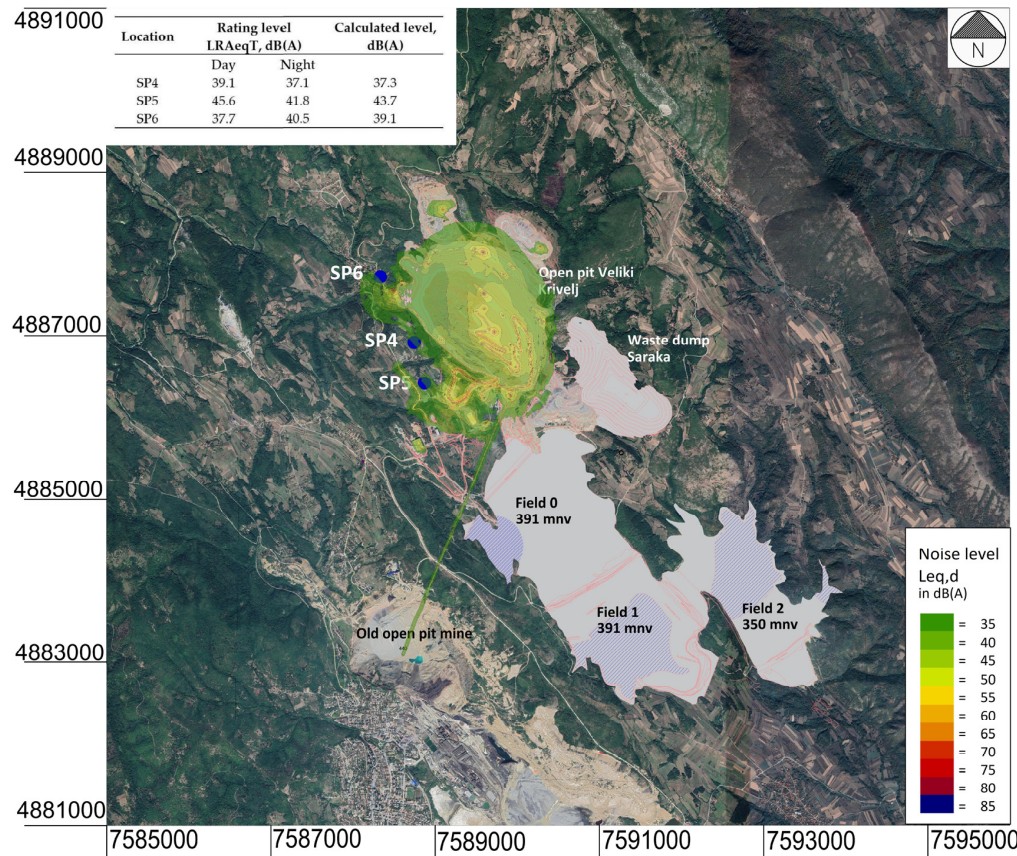

**Figure 3.** Noise levels estimation around the open-pit mine Veliki Krivelj for the model validation.

**Table 3.** Sensitive points.

| Sensitive Points | SP 1 | SP 2 | SP 3 | SP 4 | SP 5 | SP 6 |
|---|---|---|---|---|---|---|
| Building | House Trujic | Church | School | House Nikolic | House Zurkic | House Karabasevic |

As a part of the impact assessment stage in the EIA process, noise modeling was done. Noise modeling results in the above scenario are shown below in Figure 4. It can be concluded that a negative noise impact should be expected at SP 4, outside the building, given the fact that the noise level values within the zone of SP4 will exceed the permissible noise level of 55 dB(A) allowed during the day and evening (Figure 4). However, the proximity of this residential building to the open-pit mining area boundaries certainly indicates the necessity of considering its relocation, from multiple aspects, and not only due to the unfavorable noise influence (suspended particles impact, seismic effects from blasting, etc.). The residential buildings in the village will not be affected by increased levels during the day and evening.

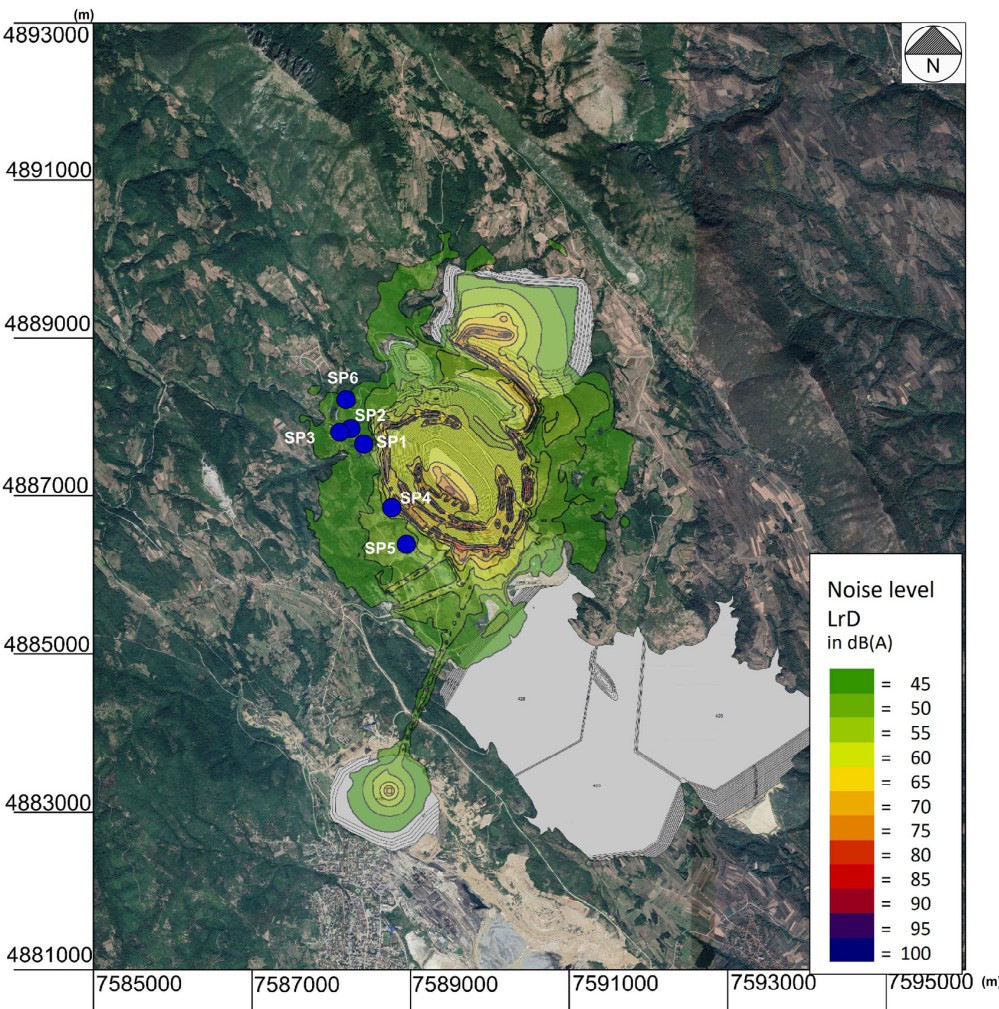

**Figure 4.** Noise levels estimation around the open-pit mine Veliki Krivelj for the above scenario with the locations of sensitive points SP1–SP6.

Regarding the noise level during the night (Figure 4), outside the building, it is estimated that the impact zone, due to the terrain configuration, will expand up to 1000 m from the contour of the mine, primarily towards the west, southwest, and south, relative to the western contour of the mine. Hence, a number of residential buildings, located west–southwest of the western contour of the mine, could be affected by the increased noise levels at night. This exposure is a consequence of the terrain configuration, i.e., the elevation of the terrain in the direction of residential buildings, which makes them much more exposed to noise originating from the open-pit mine. At the same time, the village can almost entirely be affected by unfavorable noise levels at night since the forecast noise levels are close to the limit of 45 dB(A) and any possible noise increase at the mine would lead to an increase in noise in the residential area.

Following the assessment of noise impact, it has been possible to rank the on-site noise sources depending on their contribution to the specific sound level at each receptor, as presented in Table 4.

**Table 4.** Ranking of noise sources.

| | Source Contribution | LwA * dB(A) | | Contribution Level | LwA * dB(A) |
|---|---|---|---|---|---|
| SP1—House Trujic | Dump trucks—on overburden | 36.5 | SP4—House Nikolic | Drilling rig | 57.2 |
| | Drilling rig | 36.3 | | Dump trucks—on overburden | 50.9 |
| | Dump trucks—on ore | 34.7 | | Dump trucks—on ore | 49.2 |
| SP2—Church | Dump trucks—on overburden | 38.9 | SP5—House Zurkic | Ore conveyor belt | 48.6 |
| | Drilling rig | 38.8 | | Dump trucks—on ore | 45.3 |
| | Dump trucks—on ore | 38.6 | | Dump trucks—on overburden | 44.8 |
| SP3—School | Dump trucks—on overburden | 39.1 | SP6—House Karabasevic | Drilling rig | 38.1 |
| | Drilling rig | 38.8 | | Dump trucks—on overburden | 37.1 |
| | Dump trucks—on ore | 36.9 | | Dump trucks—on ore | 35.2 |

* LwA—assessed noise level without noise control measures.

As each receptor is located at a different distance and direction from the site, the contributing noise sources will also be different depending on their relative position within the site.

Frequently, the assessment of impacts will reveal damaging effects on the environment. These may be alleviated by mitigation measures. Mitigation involves taking measures to reduce or remove environmental impacts. For example, the successful design of mitigation measures could possibly result in the removal of all significant impacts had the mitigation measures been included from the start in an early phase of planning. Mitigation of noise originating from mining machinery, equipment, and related processes, concerning a specific object or the presented technology and spatial relationship of the open-pit mine Veliki Krivelj and residential buildings, could be done by reducing the number of individual noise sources (trucks, bulldozers, graders, crushers, etc.); changing work organization (spatial and temporal) in order to disperse individual noise sources, i.e., displacing them during certain periods of the day, evening, or night, from the most sensitive objects and by the relocation of sensitive receptors significantly impacted by noise originating from the open-pit mine. Considering any of these possibilities during the production phase would have unforeseeable consequences in terms of the efficiency and economy of the entire project. The main challenge in such cases is to organize the work of the mobile equipment engaged in the mining process so that would provide the planned capacities while respecting the restrictions regarding the permitted noise levels from the equipment and blasting process. It is not possible just to reduce the amount of equipment and move them further away.

This issue is further complicated by the lack of additional technical solutions regarding noise reduction at the mobile sources. Changes in the mining management plan could be one of the possible solutions which need confirmation at the earliest stage of the mine design. Such issues require noise modeling during the project planning phase.

The temporal and spatial work schedule on the open-pit mine is the most realistic solution and should secure the planned production with minimal disturbance to the environmental quality regarding noise. Several options which corresponded to the different spatial distribution of mining activities were modeled to obtain the required noise level in the zone of the nearest residential buildings. Figure 5 shows such a case.

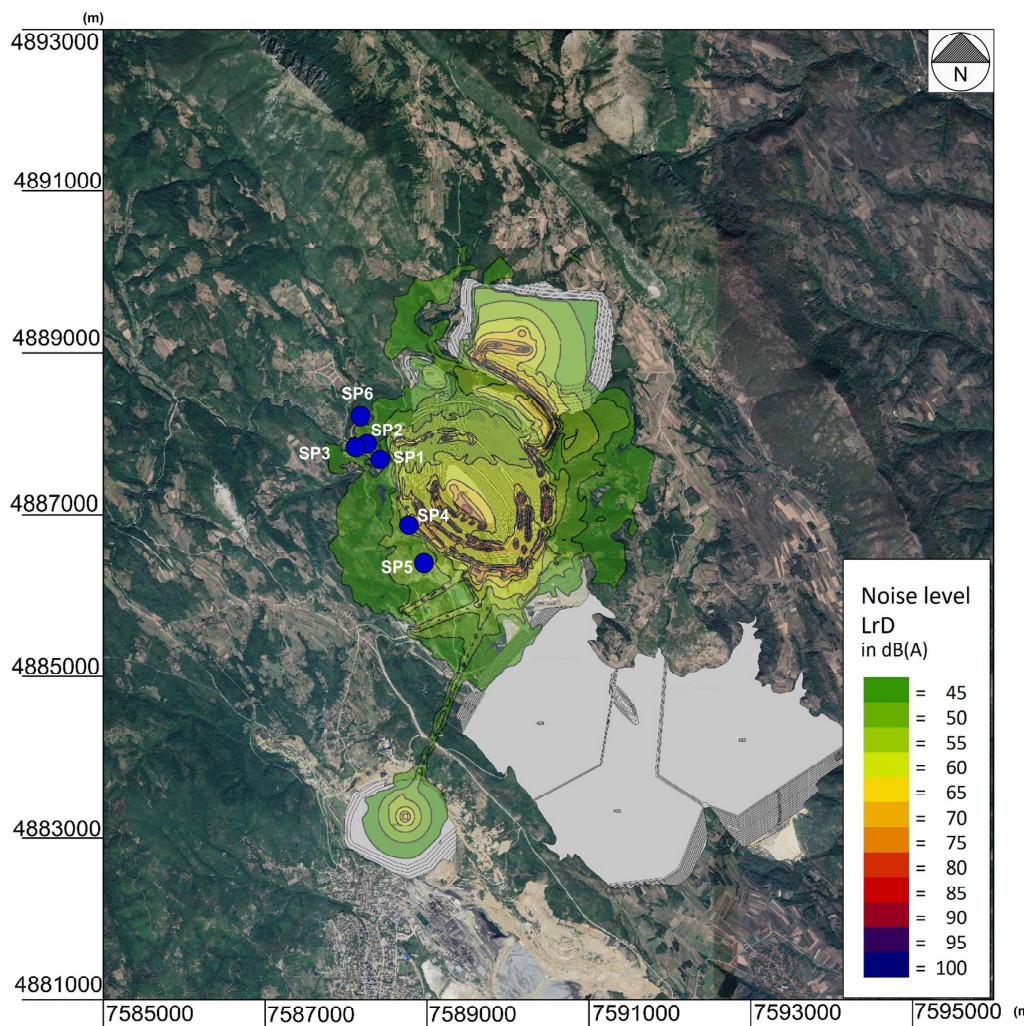

**Figure 5.** The night noise level estimation on sensitive points SP1–SP6 and around the open-pit mine Veliki Krivelj, resulting from modeling different spatial organization of work in the mine.

Based on the presented model (Figure 5), it can be concluded that it is possible, in given conditions, to establish such work conditions using different spatial layouts so as not to disturb the planned production rate while meeting the requirements regarding the permitted noise levels at night, in the immediate vicinity of the mine, primarily in the Krivelj village.

The noise level at the border of the open-pit mine, in the zone of the nearest residential buildings in the village, must not be higher than 50 dB(A), during night work. This conclusion enables proactive noise management at the open-pit mine by setting up a monitoring station to measure the noise level at the mine's border, in the zone closest to the residential buildings of the village. A real-time noise level monitoring system could be paired with appropriate action plans, in terms of spatial reorganization of work, to support the predefined noise levels at the edge of the mine, not only during the night but also during the day and evening. This would enable adequate real-time responses at any given time concerning noise issues, by a simple reorganization of activities, in accordance with the defined action plans, which is the goal of proactive noise management [45].

*3.2. Assessment and Control of Noise Induced during Blasting Operations in the Open-Pit Mine*

Most rocks require blasting before excavation. Blasting is one of, if not the most important, technological operation in most mines, bearing in mind that if it is not done successfully, the mine's sustainability is often compromised [46]. The main factors influencing

blasting results are the properties of the explosives, the number of explosives, the overall geometry of the minefield, as well as the structure of the blasted rocks.

Airblasts are the type of blasting that the population complains about the most because they manifest as sudden, unpleasant, and even frightening sound surges. If they are of high intensity, in addition to disturbing the population, they can also result in serious consequences to the hearing organs, and in certain cases, they can also cause damage to buildings. An airblast is a disturbance of pressure that spreads through the air like any other sound and is quantified in the same way as any noisy event. Due to the impulsive nature of the explosion, these manifestations are usually called "overpressure" (a temporary increase in the pressure of the surrounding air in relation to the existing atmospheric pressure). The overpressure of air, generally measured in decibels using the linear frequency weighting (linear scale (Z) or unfiltered), is expressed in decibels (dB), or in pascals (Pa) when the SI (metric) system is used. The effects of overpressure on people and objects in the environment are shown in Table 5 [47].

**Table 5.** Typical overpressure effects on people and objects.

| Pa | dB | Typical Effects |
|---|---|---|
| 20,000 | 180 | Construction damage |
| 12,620 | 176 | Wall mortar cracking |
| 6325 | 170 | Most windows broken |
| 632 | 150 | 1% of windows broken |
| 200 | 140 | No windows broken |
| 21 | 120 | Headache caused by continuous sound |
| 14 | 117 | Window glass vibration |
| 2.1 | 100 | Pneumatic hammer sounds |
| 0.02 | 60 | Normal speech |
| $2 \times 10^{-5}$ | 0 | Audibility limit |

The allowed sound levels globally are limited to the range of 120–140 dB, depending on the detonation frequency. The national regulation "Technical norms manual on handling explosive devices and blasting in the mining industry", Official Gazette No. 26/1988, also determines the permissible sound level as a function of the frequency of detonations ranging from 1 mbar (134 dB) up to 5 mbar (148 dB), which is greater than the allowed levels in other countries with developed mining industries. Considering global experiences, the recommended airblast limit values are shown in Table 6 [48].

**Table 6.** Recommended airblast overpressure limit values.

| Detonation Frequency | Maximum Pressure Increase | |
|---|---|---|
| | (Pa) | (dB) |
| Multiple detonations per day | 21 | 120 |
| Multiple detonations up to twice a week | 100 | 134 |
| Up to two detonations per week or fewer | 200 | 140 |

Table 7 provides the blasting effects assessment criteria concerning airblast, i.e., overpressure, used in some countries around the world (Canada (Ontario), the USA, Australia, and the UK).

**Table 7.** Blasting effects assessment criteria from the point of view of overpressure airblast.

| Emission Type | Receptor | Regional Criteria | | | |
| --- | --- | --- | --- | --- | --- |
| | | Ontario | USA | Australia | UK |
| Airblast dB(Z) | Residential | 128 [a] | 129 (<6 Hz) [b] 133 (<2 Hz) [b] 134 (<0.1 Hz) [b] | 115 (95%) [c] 120 (max) [c] | 120 [d] 120 (95%) [e] 125 (max) [e] |

[a] Ontario Limits for Quarries (Canada); [b] Office of Surface Mining Reclamation and Enforcement (OSMRE); [c] Australian and New Zealand Environment and Conservation Council (ANZECC) Technical Basis for Guidelines to Minimise Annoyance due to Blasting Overpressure and Ground Vibration (ANZECC, 1990); [d] British Standard BS5228 (2009) Code of Practice for noise and vibration control on construction and open sites; [e] Minerals Technical Advice Note 2: Coal (MTAN), January 2009 (Wales).

Summarizing the criteria shown in Table 7, the RioTinto company developed the criteria for its project in Guinea [49] as the basis on which it is possible to qualitatively assess the level of airblast overpressure impact in the residential zone. The criteria are shown in Table 8. The criteria are given as 95-percentile values. This is just another in a series of criteria worldwide for the assessment of blasting effects.

**Table 8.** Criteria for evaluation of impacts from blasting.

| Period | Airblast dB(Z) 95-Percentile | | | |
| --- | --- | --- | --- | --- |
| | Not Significant | Minor/ Moderate | Major | Critical |
| Daytime | <115 | >115–125 | >125–140 | >140 |
| Nighttime | <105 | >105–115 | >115–140 | >140 |

It is obvious that there are slight differences between all the criteria. While the criteria from Table 7 are maximum airblast overpressure oriented, the criteria in Table 8 are much more suitable when we have to access the possible risk from blasting regarding the airblast overpressure. In other words, there is not only one criterion widely adopted in the mining industry.

The level of overpressure is related to the number of explosives initiated at the given moment and the distance from the place of blasting. The overpressure prediction, for the purposes of this research, was done using the equation proposed by McKenzie in 1993 [50,51], in order to relate the decrease of overpressure levels as the distance from the place of explosive initiation increases:

$$OP = K - c \cdot \log_{10}\left(\frac{D}{\sqrt[3]{W}}\right) \qquad (1)$$

where:

- $OP$ is the overpressure level, read as a linear instrument response, without frequency weighting, in dB;
- $W$ is the maximum instantaneous charge initiated (per single delay), in kg;
- $D$ is the distance from the place of blasting, in m;
- $K, c$ are the constants depending on the specific blasting conditions [50] (related to the category of parameters which is influenced by design parameters including charge weight, distance from the explosion source, charge diameters, delay interval, burden, spacing, subdrilling, etc.). For practical use, a value of 165 is used for "$K$" and a value of 24 for "$c$".

Hence, Equation (1) becomes:

$$OP = 165 - 24 \cdot \log_{10}\left(\frac{D}{\sqrt[3]{W}}\right) \qquad (2)$$

Two types of explosives are used at the open-pit mine Veliki Krivelj, ANFO and Slurry, depending on the characteristics of the blasted rocks. The quantities of instantaneous charge are:

- for ANFO, 504 kg;
- for slurry, 694 kg.

According to these amounts, the levels of overpressure that can be expected in the residential buildings zone of the village (sensitive points SP1, SP2, and SP3) will depend on the specific distances of the sensitive points to the blasting site. Sensitive point 1 (SP1) is closest to the boundary of the mine and as such is significantly impacted by airblasts. The shortest expected distance to the contour of the mine, i.e., to the location of the possible blasting, will be 290 m. At the same time, the distance of sensitive point 2 (SP2) will be 369 m, and the distance of sensitive point 3 (SP3) to the place of explosives initiation will be 500 m. For the given conditions, the overpressure levels that can be expected at a given moment are given in Table 9, as well as the level of impact according to the criteria for evaluation of impact from blasting shown in Table 8.

**Table 9.** The overpressure levels at the sensitive points with the estimated level of impact according to the criteria shown in Table 7.

| Measuring Point (D, m) | Overpressure Levels (dB(Z)) $OP = 165 - 24 \cdot \log\left(\frac{D}{\sqrt[3]{W}}\right)$ | |
|---|---|---|
| | ANFO (W = 504 kg) | Slurry (W = 694 kg) |
| SP1 (290 m) | 128 | 129 |
| SP2 (369 m) | 125 | 126 |
| SP3 (500 m) | 122 | 123 |

As per data in Table 9 and according to the national regulatory recommendations (Table 6), the overpressure is above the allowed limit (120 dB) at all sensitive points, for the planned schedule of blasting, which includes one blasting per day during the day period.

Figure 6 shows the safe distance limit according to the national recommendation, which for a given value of 120 dB(Z) is 600 m from the boundary of the mine. The listed sensitive points (SP1–SP3) are exposed to a minor/moderate to a major impact, depending on the explosive used. Only daily risk is accounted for since no night blasting is planned.

Figure 7 shows the range of individual impact level zones, according to the criteria given in Table 7, relative to the contour of the mine (impact without significance: <115 dB(Z) (>1000 m); minor/moderate impact: >115–125 dB(Z); major impact: >125–140 dB(Z) (90 m); critical impact: >140 dB(Z) (<90 m).

Airblast overpressure is difficult to regulate because it differs greatly in the way it occurs and spreads and how it affects people and structures. Extremely high air pressure values can sometimes occur far from the open-pit mine and can cover significantly larger areas than are usually associated with soil vibrations. Additionally, weather conditions can contribute to airblast propagation through focusing caused by temperature inversions and wind [51]. Because of that, it is highly important to take all available mitigation measures in order to reduce the airblast overpressure level as much as possible.

Mitigation measures aimed at reducing the impact of airblast overpressures during the blasting process can be summarized as follows: avoiding explosive detonation on the rock surface; using proper stemming; if possible, planning the blast holes in such a way as to direct the blast in the opposite way of the village or the protected object (sensitive points); limiting the number of explosives for simultaneous initiation (amount of explosives per single delay); avoiding blasting during strong winds in the direction of buildings or the village; carrying out blasting according to the planned schedule and, if necessary, inform the local community appropriately, in advance; establishing an airblast overpressure level monitoring system and use the results to inform the local community as well as for future blasting projects.

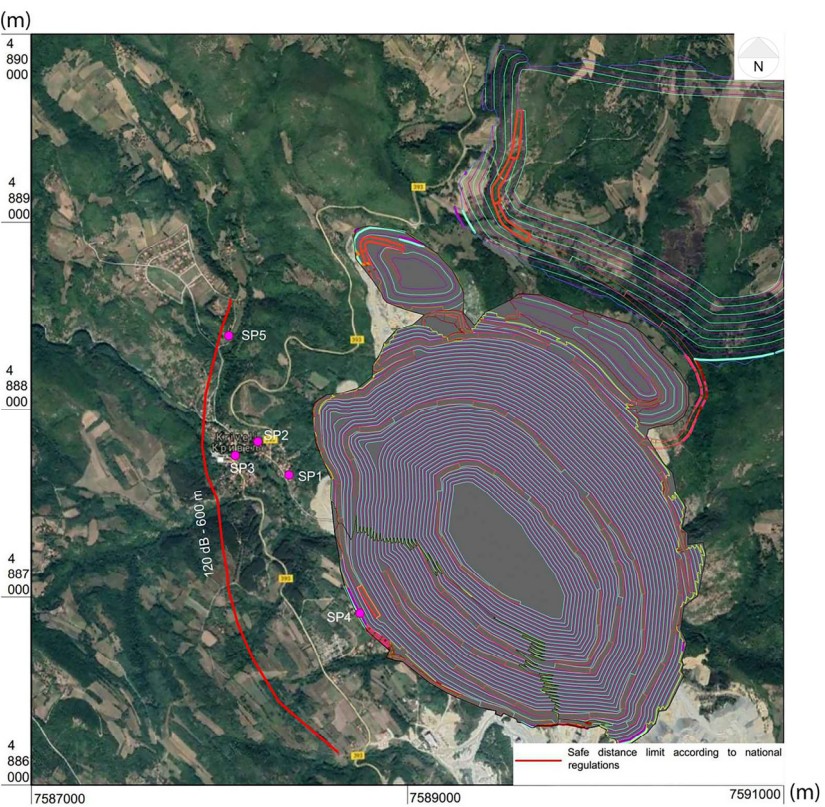

**Figure 6.** Safe distance limit according to national regulations (Table 4).

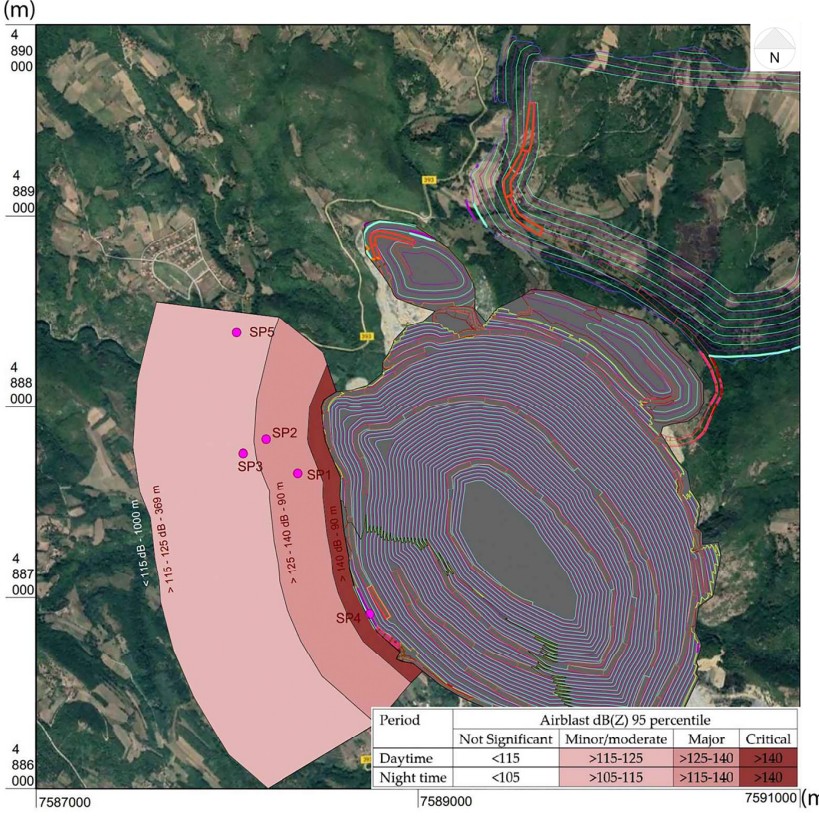

**Figure 7.** Impact levels zones as per RioTinto criteria (Table 7) for the day period.

Assuming that all the above measures have been complied with, we analyzed a case study of blasting that requires the reduction of the simultaneously initiated amount of explosives (per one delay), for the given conditions of blasting and spatial relations of the mine and residential buildings. The calculation of the required instantaneous charge per delay (in kg), in order to reduce the overpressure level to 120 dB(Z) (as recommended by national regulations) around the nearest residential building (SP1), will be done by applying Equation (2):

$$OP = 165 - 24 \cdot \log_{10} \left( \frac{D}{\sqrt[3]{W}} \right),$$

$$120 = 165 - 24 \cdot \log_{10} \left( \frac{290}{\sqrt[3]{W}} \right),$$

i.e., W = 58 kg

Based on the results, it can be concluded that in the nearest residential building zone, the maximum required instantaneous charge should not exceed 58 kg, regardless of the type of explosives used. As the mining operations could not achieve the required working efficiency with constraints regarding the estimated instantaneous loads of 58 kg, the relocation of the village is imposed as a unique solution.

As previously stated, the mining company Serbia Zijin Copper DOO Bor (Bor town, Eastern Serbia) developed a new mine plan for the open-pit mine Veliki Krivelj predicting the increase of the annual ore production rate from the existing 10.6 million TPA (tons per annum) to 23.1 million TPA of dry ore. After the environmental impact assessment of the planned and designed mining operations, the company initiated a discussion with the residents of the Krivelj settlement about the possible relocation of the village. Residents accepted relocation as a proposed option, but the discussion escalated in the direction of different modalities of compensation due to the wide range of individual demands.

## 4. Conclusions

The identification and assessment of noise impact, as well as the control of its propagation, are part of an integrated and holistic process of impact assessment and management, with a large number of interactive and competitive activities. In line with the systematic approach to impact identification and assessment, the environmental management and impact assessment system can provide a focus on significant impacts, identifying them at an early stage of project planning.

The study analysis presented in this paper focuses on the methodology and tools that can assist in the environmental noise impact assessment of large-scale surface mining operations. It follows a phased approach, starting with the definition of the noise study domain boundaries, continuing with the identification and discussion of relevant noise sources and recognition of the noise-sensitive receptors, and concluding by setting a principle-based approach to mining area noise mapping. Concerning the assessment of noise impact, it has been possible to rank the on-site noise sources depending on their contribution to the specific sound level at each receptor. This provides a basis for deciding whether noise management strategies are feasible.

The management of noise emissions during mine planning is a complex procedure, due to the numerous parameters that influence the emission and the propagation of noise. In such situations, the application of a noise propagation prediction model is currently the best practice in noise management.

According to the proposed approach, the modeling software SoundPlan 8.1 was used for creating noise maps, based on noise emissions originating from the operating mining machinery and equipment, including the processing activities.

In the case of large-scale surface mining operations, the application of the main options for reducing the noise distribution, i.e., reducing noise at the sources, along the paths of noise propagation, or at receivers, has a limited application, mostly depending on a case-

by-case basis. The proposed conceptual framework for the integration of noise impact assessment and control in the planning phase of the environmental management system provides a focus on significant impacts, thus facilitating their identification at an early stage in long and short-term mine planning.

One of the main challenges in open-pit truck haulage system design is to organize the work of the mobile equipment engaged in the mining process so that would provide the planned capacities while respecting the restrictions regarding the permitted noise levels from the equipment and other mining activities. It is not possible just to reduce the amount of equipment and move them further away. Changes in the mining management plan could be one of the possible solutions which need confirmation at the earliest stage of the mine design. Such issues require noise modeling during the project planning phase. The temporal and spatial work schedule on the open-pit mine is imposed as the most realistic solution, which should secure the planned production with minimal disturbance to the environmental quality regarding noise.

Determining the airblast overpressure impact significance and level (noise in the range of infrasound) as a consequence of the rock blasting within the subject project was done using a generally accepted approach and criteria for assessing the impact levels.

The research conducted during the development of the Serbia Zijin Copper DOO Bor mine project included an analysis and a proposal of measures to mitigate their impact in addition to an assessment of possible impacts of mining operations on the quality of the environment. Therefore, the modeling and prediction of noise propagation, i.e., airblast overpressure, were successfully implemented in assessing the efficiency of the proposed measures within the project's environment.

The environmental management approach paired with the environmental impact assessment as presented in this paper and properly implemented in the planning phase of a mining project could make the difference between an economically feasible project and its economic failure. Implementing a proactive environmental management system can improve productivity and community relations and hence generate more profit for mines.

**Author Contributions:** U.P., writing—review and editing of the manuscript and provided methodology; P.L. provided data statistics and model analysis; A.C., provided visualization and results of statistical analysis and the software; N.L., provided the funding acquisition and writing—review of the manuscript. All authors have read and agreed to the published version of the manuscript.

**Funding:** This research received no external funding.

**Institutional Review Board Statement:** Not applicable.

**Informed Consent Statement:** Not applicable.

**Data Availability Statement:** Not applicable.

**Acknowledgments:** The research described in this paper was performed during the development of the Feasibility study of Veliki Krivelj mineral deposit exploitation. The development of this project is financed by Serbia Zijin Copper DOO Bor, 2021.

**Conflicts of Interest:** The authors declare no conflict of interest. The funders had no role in the design of the study; in the collection, analyses, or interpretation of data; in the writing of the manuscript, or in the decision to publish the results.

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
