# Peer review of "Environmental Noise Impact Assessment for Large-Scale Surface Mining Operations in Serbia"

_sustainability, doi:10.3390/su15031798_

Round 1
Reviewer 1 Report
The paper is one of few that are investigating noise in mines. For these reasons, I believe that maximum attention should be put by authors in the present work, as it can become a reference for many readers in the future. By the way, the present status is not like that, and correction should be made to reach excellence. First of all, the paper should provide more general results to be used by others, and being presented in a different way, and not just like a case study.
Avoid brands.
Please report references in the right order.
Always check for the right use of dB(A) or dB.
Always put a space between units and digits.
Conclusions are not clear. They should be better summarizing the results, in order to make them more useful to other researchers. In fact, at present the results themselves looks more like a simple case study, then something that other can read. I suggest to the authors to re-edit a bit the text toward the aim of producing results that can be used by others. Otherwise, the present status maybe not interesting for readers.
Reviewer 2 Report
L 185 – Change: "are including" to: "include".
Table 1 and whole text - Would be good to use "maximum permissible level" instead of “maximal allowed level", as it is e.g. in L 202.
L 221 - 3 Results - Would be good to specify what results are in the title - "Results of ...";
Eventually I suggest “Results and discussion“.
Figure 3 and other – Please, specify the meaning of the numbers on x and y axes.
L 287 - keep together "1000 m".
Table 4 - the values in Pa are not correct: for 140 dB it should be 200 Pa, for 150 dB 632 Pa, for 170 dB 6325 Pa, for 176 dB 12620, for 180 dB 20000 Pa.
L 483 - 4 CONCLUSION - the title should not be in capital letters, correct to: "Conclusions".
Reviewer 3 Report
The aim of this paper is assessing the environmental noise impact for large-scale surface mining operations in Serbia. Particularly, the case of Serbia Zijin Copper DOO Bor mine is examined.
The paper is quite interesting and satisfactorily structured, while the majority of the most significant sections (Introduction, Materials and Methods, Results, Conclusions) have been considered. Furthermore, the “Materials and Methods” and “Results” sections are divided into several sub-sections, providing additional information. Regarding the mathematical analysis in the “Results” section, it is accurate and consistent with the extracted findings. However, some corrections should be implemented, which will lead to the overall paper improvement:
Lines 29-34: Although the first paragraph of the “Introduction” section is theoretical, no bibliographic references have been cited. In particular, some indicative, recent and relative papers, which can be optionally cited are the following: 1. Chi, M., Li, Q., Cao, Z., Fang, J., Wu, B., Zhang, Y., Wei, S., Liu, X., & Yang, Y. (2022). Evaluation of water resources carrying capacity in ecologically fragile mining areas under the influence of underground reservoirs in coal mines. Journal of Cleaner Production, 379, 134449. https://doi.org/10.1016/j.jclepro.2022.134449, 2. Castillo, G., Brereton, D., 2018. Large‐scale mining, spatial mobility, place‐making and development in the Peruvian Andes. Sustainable Development 26, 461–470. https://doi.org/10.1002/sd.1891, 3. Karagianni, A., Lazos, I., & Chatzipetros, A. (2019). Remote Sensing Techniques in Disaster Management: Amynteon Mine Landslides, Greece. In Lecture Notes in Geoinformation and Cartography. https://doi.org/10.1007/978-3-030-05330-7_9, 4. Hilson, G., 2019. Why is there a large-scale mining ‘bias’ in sub-Saharan Africa? Land use policy 81, 852–861. https://doi.org/10.1016/j.landusepol.2017.02.013. Please, include relative references, in order to bibliographically support this paragraph.
Lines 102-111: These two paragraphs explain the objective of the paper. Please, merge them into one paragraph.
Line 182: The Figure 2 caption is brief and includes limited information. Moreover, SP1-SP6 are not defined in a legend or the caption. Please, modify.
Line 221: A “Discussion” section is missing from the paper. However, a satisfactory discussion is performed in the “Results” section. Therefore, I suggest renaming “Results” section into “Results-Discussion”. Please, apply.
Line 255: The Figure 5 resolution is poor, and its greatest part is blur. Please, increase the resolution.
Line 284: Similarly to the Figure 2 caption. Please, modify.
Line 341: Similarly to the Figure 2 caption. Please, modify.
Line 436: Please, modify “Conclusions” section. In the current form, it includes information that has already been mentioned in the abstract and the “Introduction” section. Please, remove the unnecessary information and focus on the major findings of the paper. Maybe, numbering of the conclusion remarks could be applied.
Round 2
Reviewer 1 Report
The authors did a great job understanding my suggestions and applying them all over the document.
In my opinion, the paper is now ready for being accepted.
Author Response
Dear reviewer,
thanks for the suggestions given, we think they improved our workpaper and thanks for the comments on the workpaper after the corrections.
Best regards